# Toripalimab plus capecitabine in the treatment of patients with residual nasopharyngeal carcinoma: a single-arm phase 2 trial

Patients with residual nasopharyngeal carcinoma after receiving definitive treatment have poor prognoses. Although immune checkpoint therapies have achieved breakthroughs for treating recurrent and metastatic nasopharyngeal carcinoma, none of these strategies have been assessed for treating residual nasopharyngeal carcinoma. In this single-arm, phase 2 trial, we aimed to evaluate the antitumor efficacy and safety of toripalimab (anti-PD1 antibody) plus capecitabine in patients with residual nasopharyngeal carcinoma after definitive treatment (ChiCTR1900023710). Primary endpoint of this trial was the objective response rate assessed according to RECIST (version 1.1). Secondary endpoints included complete response rate, disease control rate, duration of response, progression-free survival, safety profile, and treatment compliance. Between June 1, 2020, and May 31, 2021, 23 patients were recruited and received six cycles of toripalimab plus capecitabine every 3 weeks. In efficacy analyses, 13 patients (56.5%) had complete response, and 9 patients (39.1%) had partial response, with an objective response rate of 95.7% (95% CI 78.1-99.9). The trial met its prespecified primary endpoint. In safety analyses, 21 of (91.3%) 23 patients had treatment-related adverse events. The most frequently reported adverse event was hand-foot syndrome (11 patients [47.8%]). The most common grade 3 adverse event was hand-foot syndrome (two patients [8.7%]). No grades 4-5 treatment-related adverse events were recorded. This phase 2 trial shows that combining toripalimab with capecitabine has promising antitumour activity and a manageable safety profile for patients with residual nasopharyngeal carcinoma.

Nasopharyngeal carcinoma is an epithelial cancer that develops from the mucosal lining of the nasopharynx[1,2]. The geographic distribution of nasopharyngeal carcinoma is markedly uneven; over 70% of new cases occur in southern China, southeast Asia, and north Africa[1,2]. More than 70% of patients with nasopharyngeal carcinoma are classified as having locoregionally advanced disease, which is associated with unfavourable survival outcomes[3]. Over the past few decades, much efforts have been made to improve the locoregional and distant control of this disease through photon-based radiotherapy techniques and the combination of chemotherapy and radiotherapy[1]. Nevertheless, although most patients achieve complete response after standard-of-care treatment, residual disease occurs in approximately 6.8–13% of

✉ e-mail: lvxing@sysucc.org.cn

patients, in either the nasopharynx or regional lymph nodes or both[4–7]. Previous studies found that residual disease was a negative prognostic factor, contributing to poor survival[4,8,9]. Thus, aggressive treatments of patients with residual nasopharyngeal carcinoma are crucial. The most common therapy for residual nasopharyngeal carcinoma is re-irradiation because the precise radiation dose can be conveniently applied to the nasopharynx and/or regional lymph nodes[1,10,11]. However, the overall incidence of re-irradiation-related grade 3–5 toxicities were reported in the range of 16.7–33%[1,7,12]. Surgery, another treatment approach, can be used to radically remove the residual tumour and/or regional lymph nodes[1,13]. However, surgery can potentially result in severe trauma and grave complications[14–16]. Thus, novel strategies for treating residual nasopharyngeal carcinoma are urgently needed, particularly for treating candidates that is neither resectable nor suitable for reirradiation.

Immune checkpoint blockade therapy is a breakthrough in cancer treatment that can prevent tumour spread and metastasis[1,17]. The high expression of programmed death-ligand 1 (PD-L1)[18] and intense non-malignant lymphocytic infiltration[19] observed in nasopharyngeal carcinoma indicate that the potential application of immune checkpoint blockade therapy may be effective[20]. Several important trials of anti-programmed cell death-1 (PD-1) monoclonal antibodies have shown encouraging results in recurrent or metastatic nasopharyngeal carcinoma[21–25]. Toripalimab is a high-affinity, humanised immunoglo-bulin G4–κ (IgG4–κ), monoclonal antibody that specifically binds to PD-1[21,22,26]. It has shown promising clinical efficacy and favourable safety with manageable treatment compliance in several clinical trials involving patients with recurrent or metastatic nasopharyngeal carcinoma[21,22]. However, to date, this drug has not been assessed for patients with residual nasopharyngeal carcinoma.

Capecitabine, an oral prodrug of fluorouracil, inhibits cell division and interferes with RNA and protein synthesis. A number of studies have reported that capecitabine shows encouraging treatment efficacy and tolerable toxicities in patients with recurrent or metastatic nasopharyngeal carcinoma[27,28]. Furthermore, in a phase 3 trial, Chen and colleagues found that the addition of metronomic adjuvant capecitabine to chemoradiotherapy significantly improved survival in patients with locoregionally advanced nasopharyngeal carcinoma[29]. In another phase 2 trial by Miao and colleagues, adjuvant capecitabine following concurrent chemoradiotherapy was well tolerated and improved failure-free survival among patients with locoregionally advanced nasopharyngeal carcinoma[30]. However, in patients with residual nasopharyngeal carcinoma, the efficacy and safety profile of capecitabine is unclear.

In this trial, we present the results of a phase 2 trial assessing the antitumor efficacy and safety of toripalimab plus capecitabine for patients with residual nasopharyngeal carcinoma after definitive treatment.

## Results

Between June 1, 2020, and May 31, 2021, 25 patients were assessed for eligibility, of whom 23 patients (92%) commenced toripalimab plus capecitabine combination treatment and were included in the efficacy and safety analyses (Fig. 1). The median age of the 23 patients was 52 years (IQR 38-54), and they were mostly men (15 patients [65.2%]). Of the 23 patients, 22 patients (95.7%) received induction chemotherapy plus concurrent chemoradiotherapy. The most common locations of the residual nasopharyngeal carcinoma were nasal skull base (five patients [21.7%]) and cervical lymph nodes (five patients [21.7%]). The detectable and undetectable plasma Epstein–Barr virus DNA were obtained in 2 patients (8.7%) and 21 patients (91.3%), respectively. Ten patients (43.5%) had pathological or cytological diagnoses; and 13 patients (56.5%) had radiological diagnoses. The baseline patient characteristics are summarised in Table 1. The details of administration of upfront chemotherapy and radiotherapy are shown in Supplementary Table 1–5.

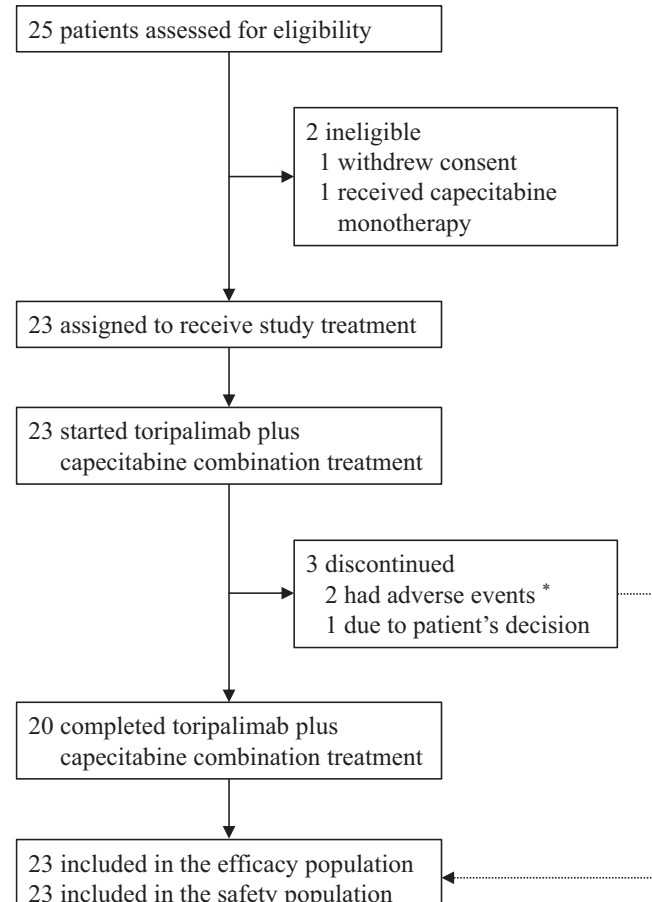

**Fig. 1 | Trial profile.** *One patient permanently discontinued the study treatment after developing continuous grade 2 hypothyroidism and one patient permanently discontinued the study treatment due to grade 3 myocardial infarction.

In the efficacy population, according to RECIST version 1.1, at the end of three cycles of toripalimab plus capecitabine, four patients (17.4%) had complete response, 17 patients (73.9%) had partial response, and two patients (8.7%) had stable disease (Table 2; Fig. 2a). After the completion of six cycles of scheduled study treatment, 13 patients (56.5%) achieved complete response, nine patients (39.1%) achieved partial response, and one patient achieved stable disease (4.3%), with an objective response rate of 95.7% (95% CI 78.1-99.9), complete response rate of 56.5% (34.5-76.8), and disease control rate of 100% (95% CI 100-100) (Table 2; Fig. 2b). Median time to the best response from the treatment initiation was 4.5 months (IQR 4.2-5.1). After the completion of six cycles of toripalimab plus capecitabine, median change from the baseline was −100% (95% CI −100 to −68.8). The tumour response to treatment according to previous exposure to fluorouracil are presented in Supplementary Table 6.

After completing the study treatment, ten of 23 patients (43.5%) did not achieve complete response. Overall, five patients (50%) subsequently received treatments, including two patients (20%) who continuously received toripalimab plus capecitabine, two patients (20%) underwent surgery (converting unresectable to resectable), while the remaining patient (10%) received multidrug chemotherapy (docetaxel, cisplatin, and fluorouracil regimen). Of note, five patients (50%) underwent clinical observation only.

By the cut-off date of Sep 1, 2023, the median follow-up time was 29 months (IQR 26–33). The median duration of treatment exposure was 4.2 months (IQR 4.2–4.4). In accordance with the independent review team assessment, four (17.4% [95% CI 5-38.8]) disease progression events were documented in the 23 patients, all of which were

## Table 1 | Baseline Characteristics

| | Toripalimab plus capecitabine combination treatment (n = 23) | |
|---|---|---|
| Age, years; median (IQR) | 52 | (30–76) |
| Sex | | |
| Male | 15 | (65.2%) |
| Female | 8 | (34.8%) |
| Karnofsky Performance Status (KPS) score | | |
| 90–100 | 22 | (95.7%) |
| 80–90 | 1 | (4.3%) |
| Histology * | | |
| Differentiated non-keratinized | 1 | (4.3%) |
| Undifferentiated non-keratinized | 22 | (95.7%) |
| Primary tumor (T) category † | | |
| T2 | 1 | (4.3%) |
| T3 | 12 | (52.2%) |
| T4 | 10 | (43.5%) |
| Primary node (N) category † | | |
| N1 | 6 | (26.1%) |
| N2 | 9 | (39.1%) |
| N3 | 8 | (34.8%) |
| Primary overall stage † | | |
| II | 1 | (4.3%) |
| III | 7 | (30.4%) |
| IVA | 15 | (65.2%) |
| Induction chemotherapy | | |
| Yes | 22 | (95.7%) |
| No | 1 | (4.3%) |
| Concurrent chemoradiotherapy | 23 | (100%) |
| Location of residual disease | | |
| Nasal skull base | 5 | (21.7%) |
| Retropharyngeal lymph nodes | 2 | (8.7%) |
| Cervical lymph nodes | 5 | (21.7%) |
| Nasal skull base and retropharyngeal lymph nodes | 2 | (8.7%) |
| Nasal skull base and cervical lymph nodes | 3 | (13%) |
| Retropharyngeal and cervical lymph nodes | 2 | (8.7%) |
| Nasal skull base, retropharyngeal and cervical lymph nodes | 4 | (17.4%) |
| Plasma Epstein-Barr virus DNA § | | |
| Detectable | 2 | (8.7%) |
| Undetectable | 21 | (91.3%) |

Data are median (IQR), or n (%). Percentages (%) might not total 100% because of rounding. *IQR* interquartile range. *Histology was categorised according to the WHO Classification of Tumors. †According to the 8th edition of American Joint Committee on Cancer staging system. §A cut-off level of 40 copies per mL was used to categorised Epstein-Barr virus DNA levels.

## Table 2 | Response to The Treatment

| | Efficacy population (n = 23) | | | |
|---|---|---|---|---|
| | The end of 3 cycles of scheduled treatment | | Completion of 6 cycles of scheduled treatment | |
| Complete response | 4 | (17.4%) | 13 | (56.5%) |
| Partial response | 17 | (73.9%) | 9 | (39.1%) |
| Stable disease | 2 | (8.7%) | 1 | (4.3%) |
| Objective response rate | 21 | (91.3%; 72–98.9) | 22 | (95.7%; 78.1–99.9) |
| Complete response rate | 4 | (17.4%; 5.0–38.8) | 13 | (56.5%; 34.5–76.8) |
| Disease control rate | 23 | (100%; 100–100) | 23 | (100%; 100–100) |

Data are n (%), or n (%; 95% CI). Percentages (%) might not total 100% because of rounding. 95% CI = 95% confidence interval.

any grade (ie, those occurring in >20% of patients) included leukopenia (nine patients [39.1%]), neutropenia (eight [34.8%]), anaemia (eight [34.8%]), and lymphopenia (six [26.1%]); the most common non-haematological adverse events were hand-foot syndrome (11 patients [47.8%]), hypothyroidism (six [26.1%]), fatigue (six [26.1%]), nausea (six [26.1%]), and anorexia (five [21.7%]). Most frequently reported treatment-related grade 3 adverse event was hand-foot syndrome (two patients [8.7%]), attributed to oral capecitabine. Overall, 15 of 23 patients (65.2%) reported grade 1–2 immune-related adverse events. No patients had grade 3-5 immune-related adverse events. The most common grade 1-2 immune-related adverse events, with incidence ≥10%, included hypothyroidism (six patients [26.1%]), hyperthyroidism (three [13.0%]), fatigue (three [13.0%]), leukopenia (three [13.0%]), lymphopenia (three [13.0%]), and anaemia (three [13.0%]). Details regarding adverse events are summarised in Table 3 and Supplementary Table 7.

Regarding compliance to treatment, 20 of 23 patients (87%) completed the six cycles of toripalimab plus capecitabine combination treatment. Overall, three of 23 patients (13%) discontinued the study treatment due to adverse events (one patient [4.3%] with continuous grade 3 hypothyroidism; one patient [4.3%] with grade 3 myocardial infarction), and withdrawal of consent (one patient [4.3%]). Furthermore, 20 patients (87%) received the full dosage of toripalimab and 12 patients (52.2%) received the full dosage of capecitabine. The dose of capecitabine was reduced by one level (to 75% of the original dose) in nine patients (39.1%), and by two levels (to 50% of the original dose) in another one patient (4.3%). The reason for the dose reduction was adverse events. The median relative dose intensity for toripalimab was 100% (IQR 100–100), while that for capecitabine was 100% (89.6-100). Dose reductions of the study medications are shown in Supplementary Table 8.

## Discussion

This phase 2 trial focused on the management of residual nasopharyngeal carcinoma. This trial also enrolled the largest sample size to date for this disease of serious concern. We reported the clinical efficacy and safety of PD-1 inhibitor plus oral chemotherapy combination treatment in patients with residual nasopharyngeal carcinoma who have previously received standard-of-care treatments. Our results indicated that toripalimab plus capecitabine have favourable and durable efficacy and a manageable toxicity profile in patients with residual nasopharyngeal carcinoma.

Although nasopharyngeal carcinoma is sensitive to chemoradiation and a high proportion of patients achieve a complete response with standard-of-care treatment, the incidence of residual disease ranges from 6.8–13%[4–7]. Management of residual nasopharyngeal carcinoma is a challenge. Treatments usually consist of re-irradiation, surgery, or chemotherapy. Thus, optimisation of treatment according

locoregional recurrence. No distant metastasis was recorded. The median progression-free survival was not reached (95%CI not reached-not reached), and 12-month progression-free survival was 95.7% (95% CI 87.7–100) and 24-month progression-free survival was 82.4% (95% CI 68.1–99.7) (Fig. 3a). The median duration of response was not reached (95%CI not reached-not reached) (Fig. 3b). No deaths occurred during the study treatment and follow-up phase.

In the safety population, any grade treatment-related adverse events occurred in 21 of 23 patients (91.3%), with grade 3 adverse events in five of 23 patients (21.7%). No grade 4–5 adverse events occurred. The most common haematological adverse events of

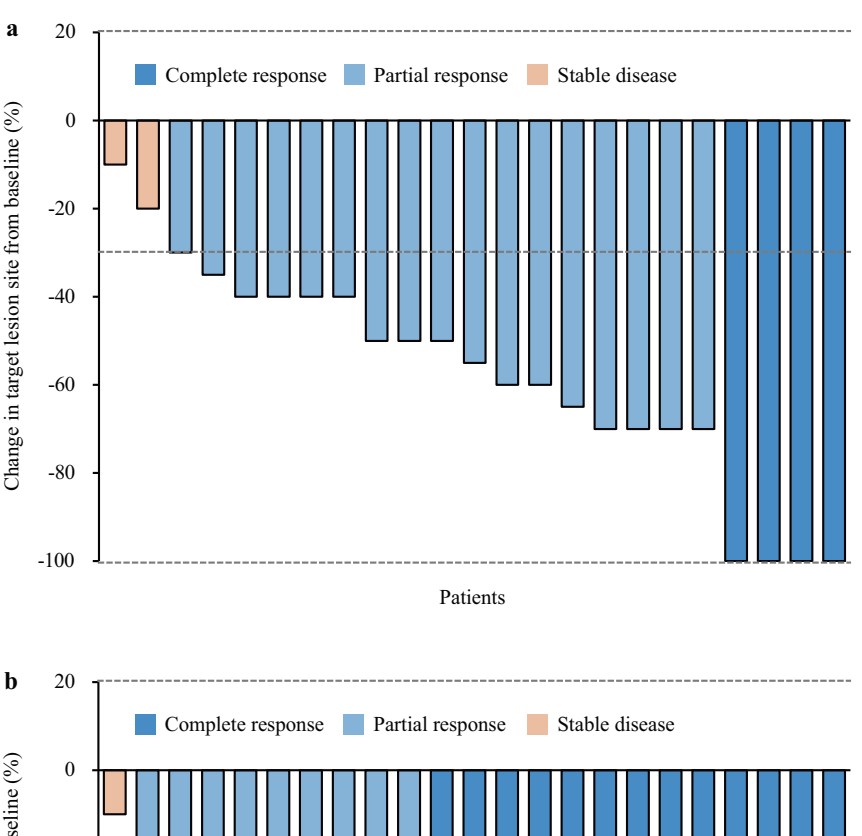

**Fig. 2 | Tumour response in the efficacy population (*n* = 23).** Waterfall plots of the best percentage change in the target lesion size from baseline to the end of the three cycles of study treatment (**a**) and from baseline to three weeks after the completion of the six cycles of study treatment (**b**). The colour indicates the type of response. The dashed line at 20% represents the boundary for the determination of progressive disease, and the dashed line at −30% represents the boundary for the determination of partial response. Source data are provided as a Source Data file.

to the patterns of residual disease has naturally attracted the most attention.

In recent years, immune checkpoint inhibitors have been extensively studied for treating nasopharyngeal carcinoma. In the POLARIS-02 study, Wang and colleagues found that toripalimab provided a durable clinical response and manageable safety profile in patients with previously treated recurrent or metastatic nasopharyngeal carcinoma[22]. A 2021 phase 3 trial (JUPITER-02) showed that the addition of toripalimab to gemcitabine plus cisplatin chemotherapy regimen for patients with recurrent or metastatic nasopharyngeal carcinoma significantly improved progression-free survival compared with the gemcitabine plus cisplatin chemotherapy regimen alone and had a manageable safety profile[21]. Based on these clinical trial results, toripalimab intervention may be a promising option for treating residual nasopharyngeal carcinoma.

Capecitabine, a convenient, orally administered fluorouracil drug, was reported to have clinical benefits in recurrence or metastatic nasopharyngeal carcinoma[27,28]. In a retrospective review, Chua and colleagues found that capecitabine monotherapy has favourable outcome for treating recurrence or metastatic nasopharyngeal carcinoma[27]. In addition, a study done by Ciuleanu and colleagues showed that patients had satisfied overall response and mild toxicity with capecitabine monotherapy in relapsed nasopharyngeal carcinoma[28]. Compared with conventional intravenous fluorouracil, capecitabine reduce toxicity without sacrificing treatment efficacy in locoregionally advanced nasopharyngeal carcinoma[31]. Considering all the available evidences, capecitabine maybe represent a promising candidate for use in residual nasopharyngeal carcinoma. In future, a series studies of capecitabine for this disease might provide more insights.

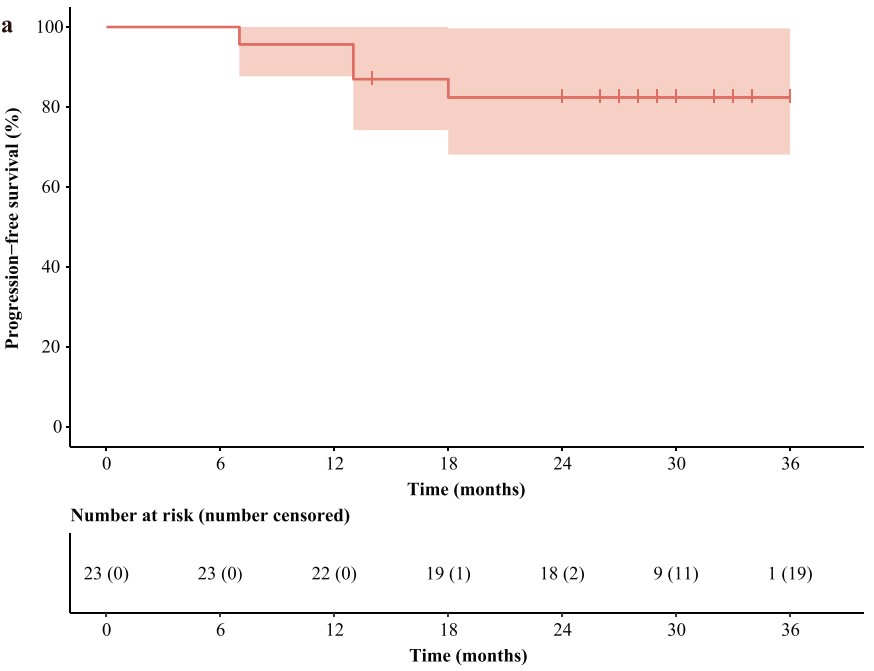

**Number at risk (number censored)**

| | | | | | | |
|---|---|---|---|---|---|---|
| 23 (0) | 23 (0) | 22 (0) | 19 (1) | 18 (2) | 9 (11) | 1 (19) |
| 0 | 6 | 12 | 18 | 24 | 30 | 36 |

Time (months)

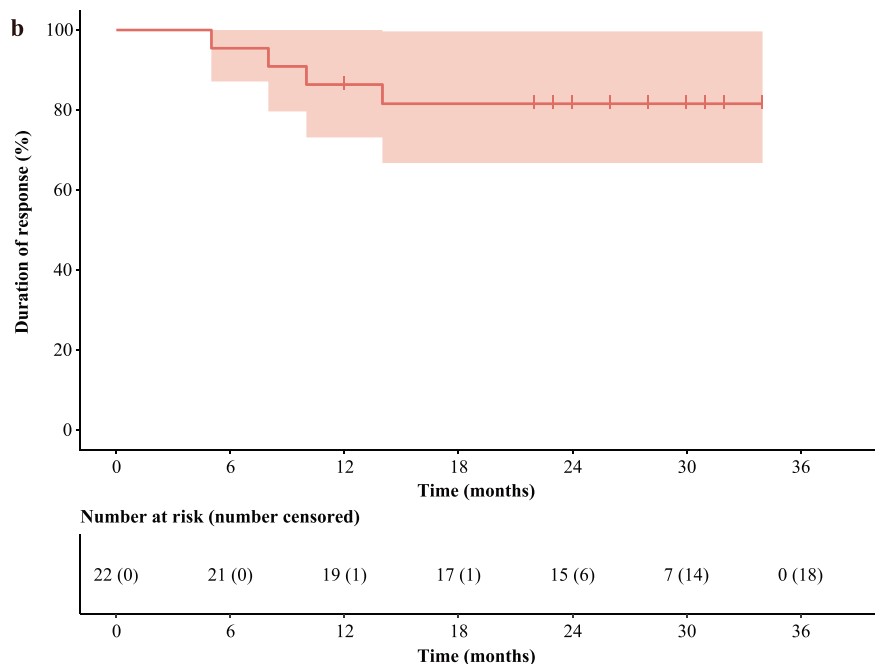

**Number at risk (number censored)**

| | | | | | | |
|---|---|---|---|---|---|---|
| 22 (0) | 21 (0) | 19 (1) | 17 (1) | 15 (6) | 7 (14) | 0 (18) |
| 0 | 6 | 12 | 18 | 24 | 30 | 36 |

Time (months)

**Fig. 3 | Kaplan–Meier analyses in the efficacy population. a** Progression-free survival in full analysis set (*n* = 23). **b** Duration of response among all patients who responded (*n* = 22). Source data are provided as a Source Data file.

Previous reports suggested that PD-1 inhibitor and oral chemotherapy which differ in mechanisms, may induce synergistic antitumour effects without the risk of overlapping toxicities[32]. Based on the results of previous preclinical studies, chemotherapy has an immunomodulatory effect that would be synergistic with anti-PD-1-based immunotherapy[33,34]. In glioblastoma, administering capecitabine as an immune modulator reduced circulating levels of myeloid-derived suppressor cells and increased cytotoxic immune infiltration into the tumour microenvironment[35]. A small phase 2 trial by Zsiros and colleagues showed that the combination of pembrolizumab with bevacizumab and oral metronomic cyclophosphamide present a promising treatment option in patients with recurrent ovarian cancer[36]. In

our trial, we found that the six-cycle administration of toripalimab plus capecitabine as a combination therapy in patients with residual nasopharyngeal carcinoma led to a proportion of 95.7% (95% CI 78.1–99.9) patients achieving an objective response and 100% (95%CI 100–100) of patients achieving disease control, including 56.5% (13 of 23 patients) complete response, 39.1% (9 of 23 patients) partial response and 4.3% (1 of 23 patients) stable disease. The exact mechanism of this effect needs further research.

Our oncological team adopts appropriate treatments to optimise clinical efficacy while constantly attempting to minimise toxicity. Poor safety and compliance with subsequent therapy after primary treatment cannot be ignored. Compared with conventional intravenous

**Table 3 | Treatment-related adverse events**

| | Toripalimab plus capecitabine treatment (n = 23) | | | | | |
|---|---|---|---|---|---|---|
| | **Any grade** | | **Grade 1–2** | | **Grade 3** | |
| Any adverse event | 21 | (91.3%) | 16 | (69.6%) | 5 | (21.7%) |
| Haematological adverse event | | | | | | |
| Leukopenia | 9 | (39.1%) | 8 | (34.8%) | 1 | (4.3%) |
| Neutropenia | 8 | (34.8%) | 7 | (30.4%) | 1 | (4.3%) |
| Anemia | 8 | (34.8%) | 8 | (34.8%) | 0 | |
| Lymphopenia | 6 | (26.1%) | 5 | (21.7%) | 1 | (4.3%) |
| Thrombocytopenia | 1 | (4.3%) | 1 | (4.3%) | 0 | |
| Non-haematological adverse event | | | | | | |
| Hand-foot syndrome | 11 | (47.8%) | 9 | (39.1%) | 2 | (8.7%) |
| Hypothyroidism | 6 | (26.1%) | 6 | (26.1%) | 0 | |
| Fatigue | 6 | (26.1%) | 6 | (26.1%) | 0 | |
| Nausea | 6 | (26.1%) | 5 | (21.7%) | 1 | (4.3%) |
| Anorexia | 5 | (21.7%) | 4 | (17.4%) | 1 | (4.3%) |
| Bilirubin increased | 4 | (17.4%) | 4 | (17.4%) | 0 | |
| Mucositis or stomatitis | 4 | (17.4%) | 4 | (17.4%) | 0 | |
| Hyperthyroidism | 3 | (13.0%) | 3 | (13.0%) | 0 | |
| Hypokalemia | 3 | (13.0%) | 3 | (13.0%) | 0 | |
| Weight loss | 3 | (13.0%) | 3 | (13.0%) | 0 | |
| Diarrhea | 3 | (13.0%) | 3 | (13.0%) | 0 | |
| Vomiting | 3 | (13.0%) | 3 | (13.0%) | 0 | |
| Sensory neuropathy | 3 | (13.0%) | 3 | (13.0%) | 0 | |
| Rash | 3 | (13.0%) | 3 | (13.0%) | 0 | |
| Alanine aminotransferase increased | 2 | (8.7%) | 2 | (8.7%) | 0 | |
| Aspartate aminotransferase increased | 2 | (8.7%) | 2 | (8.7%) | 0 | |
| Creatinine increased | 2 | (8.7%) | 2 | (8.7%) | 0 | |
| Hyponatremia | 2 | (8.7%) | 2 | (8.7%) | 0 | |
| Hypomagnesemia | 2 | (8.7%) | 2 | (8.7%) | 0 | |
| Hypoalbuminemia | 2 | (8.7%) | 2 | (8.7%) | 0 | |
| Pruritus | 2 | (8.7%) | 2 | (8.7%) | 0 | |
| Fever | 2 | (8.7%) | 2 | (8.7%) | 0 | |
| Proteinuria | 1 | (4.3%) | 1 | (4.3%) | 0 | |
| Hyperglycemia | 1 | (4.3%) | 1 | (4.3%) | 0 | |
| Myocardial infarction | 1 | (4.3%) | 0 | | 1 | (4.3%) |
| Thyroid stimulating hormone increased | 1 | (4.3%) | 1 | (4.3%) | 0 | |
| Headache | 1 | (4.3%) | 1 | (4.3%) | 0 | |
| Myalgia | 1 | (4.3%) | 1 | (4.3%) | 0 | |

Data are n (%). Some patients had more than one adverse event. No grade 4–5 adverse events were reported.

chemotherapy, oral chemotherapy may be accepted with the advantage of better patient compliance. Capecitabine is a cost-effective and easily accessible oral treatment for patients, especially for outpatients. Notably, although several clinical trials have suggested a capecitabine dose of 1250 mg/m² twice daily, on days 1–14 every 3 weeks for patients with head and neck cancers in Western countries[37,38], we modified the conventional dosage to 1000 mg/m² in our schedule. In a study by Chua and colleagues, 75.5% of patients with recurrent or metastatic nasopharyngeal carcinoma underwent a capecitabine dose reduction (from 1250 mg/m² to 1000 mg/m² twice daily) because of adverse effects[27], which suggested that most patients in China might not tolerate the conventional dosage. Considering safety and compliance, the modified capecitabine dosage was used.

The safety profile of the toripalimab plus capecitabine combination treatment observed in our study was consistent with that in previous trials[21,22]. No grade 4-5 events indicated that our study treatment was less toxic than conventional regimens. Most of the adverse events in this trial were well tolerated and manageable. The

most common grade 3 treatmentrelated adverse events was hand-foot syndrome, mainly attributed to capecitabine, which could be ameliorated by a capecitabine dose reduction. Notably, the incidence of hypothyroidism or hyperthyroidism was attributed to toripalimab; most of these adverse events were generally manageable (except for one patient who permanently discontinued the study treatment after developing continuous grade 2 hypothyroidism). One grade 3 myocardial infarction occurred, leading to hospitalisation.

Twelve weeks after completion of radiotherapy or chemoradiotherapy is widely considered the appropriate timepoint for assessing tumour response in nasopharyngeal carcinoma[4,39]. In a recent studies, Lin and colleagues reported patients with nasopharyngeal carcinoma continued to respond until three months after radiotherapy[39]. Therefore, we considered that 12–16 weeks after chemoradiotherapy was the optimal timepoint for assessing residual nasopharyngeal carcinoma in the current study.

This trial has several limitations. First, all patients were recruited from an endemic area where the most common histology of

nasopharyngeal carcinoma was WHO Type II or III; hence, its applicability outside of the endemic regions remains to be determined. Second, previous exposure to fluorouracil (in patients who received induction chemotherapy regimens containing fluorouracil) might have affected the apparent efficacy of capecitabine. However, consistent advantages were observed irrespective of chemotherapy regimens (with or without fluorouracil). Third, the plasma Epstein-Barr virus DNA load were not detectable in some patients, which may influence the judgement of residual nasopharyngeal carcinoma. Nevertheless, through summarising the incidence of detectable plasma Epstein-Barr virus DNA in published reports[40–42], we found that no obvious differences were identified. The association between the residual disease and plasma Epstein-Barr virus load remains inconclusive. Fourth, it is challenging to completely distinguish adverse events due to toripalimab from those due to capecitabine exactly. Finally, inadequate viable tumour cells in the biopsy or fine needle aspiration specimens from some patients has also hindered the analyses of predictive genetic biomarkers and underlying mechanisms.

In conclusion, our trial shows that toripalimab plus capecitabine has favourable antitumour activities and a manageable safety profile for patients with residual nasopharyngeal carcinoma who have previously received definitive treatment. Based on the results of this trial, a phase 3 trial of toripalimab plus capecitabine for residual nasopharyngeal carcinoma warrant investigation.

## Methods
### Study design and patients
This single-arm, open-label, phase 2 trial was performed at SunYat-sen University Cancer Centre in Guangzhou, China. Patients were eligible if they were 18–70 years old; had histopathologically or cytologically confirmed undifferentiated or differentiated nonkeratinising nasopharyngeal carcinoma without distant metastases (according to the 8th edition of the American Joint Committee on Cancer staging system and WHO); and had pathological, cytological, or radiological diagnosis of residual disease at 12–16 weeks after definitive treatments (concurrent platinum-based chemoradiotherapy with or without induction chemotherapy). The details of the treatments are available in the Supplementary Methods (Supplementary Information file). Other inclusion criteria were as follows: not suitable for local treatment (ie, re-irradiation, or surgery); Karnofsky Performance Status score of at least 70; and adequate haematological, renal, and hepatic function. The main exclusion criteria included: the residual disease could not be identified and measurable on magnetic resonance imaging (MRI); receiving previous chemotherapy or radiotherapy to nasopharynx or neck before definitive treatment; undergoing previous surgery (except for diagnostic procedures), biotherapy, or immunotherapy before enrolment; receiving any anti-tumour therapy for residual nasopharyngeal carcinoma before enrolment; receiving systemic corticosteroid therapy within two weeks before enrolment; having other malignant diseases; a history of active autoimmune disease; severe coexisting illness; and being pregnant or lactating. The full inclusion and exclusion criteria are presented in the study protocol (available as Supplementary Note in the Supplementary Information file).

The Institutional Ethics Review Board of Sun Yat-sen University Cancer Centre approved the trial protocol. This study was conducted in accordance with the Declaration of Helsinki and Good Clinical Practice guidelines defined by the International Conference on Harmonization. All patients provided written informed consent.

### Procedures and assessments
The first patient was enrolled on June 27, 2020, and the last patient on May 28, 2021. Essential assessments of residual nasopharyngeal carcinoma were performed within a span of two weeks before treatment initiation. These assessments included the collection of a complete medical history; physical examination; haematological and biochemical tests; urine and stool tests; thyroid function test; nasal endoscopy or rhino-sinusal endoscopy; electrocardiograms; enhanced MRI of the nasopharynx and neck; computed tomography (CT) scan of the chest; and abdominal scan (external ultrasonography or CT), bone scan, or $^{18}$F-FDG ($^{18}$F) PET-CT (if necessary). Plasma Epstein–Barr virus DNA load was performed at our institution. Biopsy or fine needle aspiration of suspected lesions was performed to confirm locally residual nasopharyngeal carcinoma. For lesions that were not accessible, the radiological diagnoses were accepted if patients exhibited at least two classic features on radiological imaging. Details on the radiological diagnostic criteria[8,39,43,44] are provided in the Supplementary Information.

Eligible patients received six cycles of toripalimab (240 mg, intravenously once daily on day 1, every 3 weeks) and capecitabine (1000 mg/m$^2$, orally twice daily on days 1–14, every 3 weeks). Treatments was continued for a maximum of six cycles or until fulfilment of a criterion for discontinuation (eg, progressive disease, intolerable toxicity, or withdrawal of consent), whichever occurred first (details are presented in the study protocol, available in the Supplementary Information). During the treatment, dose modification of toripalimab was not permitted. If grade 2 or 3 adverse events occurred, capecitabine was delayed until recovery to grade 1 or better and then resumed at the original dose or at a reduced dose (75% or 50% of the original dose). Upon the fourth occurrence of grade 2 adverse events or the third occurrence of grade 3 adverse event, capecitabine was permanently discontinued. If grade 4 adverse event occurred, capecitabine was discontinued permanently or delayed until recovery to grade 1 or better; then, it was resumed at a reduced dose (50% of the original dose). Dose modification or interruption of capecitabine due to adverse events was performed according to the protocol.

The response of residual nasopharyngeal carcinoma to the treatment was evaluated at the end of three cycles of scheduled treatment and three weeks after the completion of six cycles of the scheduled treatment. According to the Response Evaluation Criteria in Solid Tumours (RECIST; version 1.1), an independent review team evaluated the response by physical examination, nasal endoscopy or rhino-sinusal endoscopy, enhanced MRI of the nasopharynx and neck, and $^{18}$Ffluorodeoxyglucose PET-CT (if necessary). Adverse events reported by the patients were assessed, and physical examination and haematological tests were carried out on days 1 and 8 of every cycle. Biochemical tests, urine and faecal tests, thyroid function tests, electrocardiograms, and nasal endoscopy or rhino-sinusal endoscopy were performed on day 1 of every cycle. Adverse events were monitored continuously throughout the treatment period and until 60 days after the last dose of the study drugs. Adverse events were graded in accordance with the National Cancer Institute Common Terminology Criteria for Adverse Events (NCI CTC-AE; version 5.0).

One month after patients had completed or discontinued the study treatment, a follow-up visit was conducted. Then, the patients were followed up every 3 months for the first 3 years, every 6 months for the next 4–5 years, and annually thereafter. The follow-up details are specified in the protocol.

### Outcomes
The efficacy and safety were assessed by an independent review team according to RECIST (version 1.1) and NCI CTC-AE (version 5.0).

The primary endpoint of this trial was the objective response rate (three weeks after completion of six cycles of scheduled treatment), which was defined as the proportion of patients with confirmed complete or partial response according to RECIST version 1.1. The following secondary endpoints were also analysed: complete response rate (defined as the proportion of patients who had complete response); disease control rate (defined as the proportion of patients who achieved an objective response or stable disease); duration of

response (defined as the time from the first documented objective response to disease progression or death from any cause, whichever occurred first); progression-free survival (defined as the time from treatment initiation to disease progression or death from any cause, whichever occurred first); safety profile; and treatment compliance. Data for patients who had no observed events were censored at the date of the last follow-up.

## Statistical analysis

The data cut-off date for the present analysis was September 1, 2023.

The sample size was estimated according to Simon's two-stage design with a one-sided α error of 0.025 and a power of 80%[45,46]. A previous trial reported that the highest objective response rate of capecitabine monotherapy for recurrent nasopharyngeal carcinoma was 47.8%[28]. The objective response rate to the combination regimen (toripalimab plus capecitabine) was initially expected to be 80%. Under these assumptions, the stages were as follows: in stage one, among six evaluable patients, if the responders were three or fewer, the trial would be terminated. Otherwise, an additional 15 patients would be enrolled for stage two. In stage two, if 15 responders or more were observed (including those from stage one), the trial would be considered a success. Assuming a 10% dropout rate, a total of 23 patients were required for this trial.

Efficacy analyses were conducted for all assigned patients who received at least one dose of the study medications (the efficacy population). Patients who did not have at least one post-baseline efficacy assessment were excluded from the efficacy population. Safety was assessed in all assigned patients who received at least one dose of the medications in our study (the safety population). The safety population excluded patients without any safety data.

Continuous and categorical variables were expressed as the median (interquartile range [IQR]) and number (percentage [%]), respectively. We calculated the objective response rate, complete response rate, and disease control rate; and the accompanying 95% confidence intervals (95% CIs) were calculated based on the Clopper-Pearson method. The median duration of response and median progression-free survival were estimated using the Kaplan–Meier method, and the corresponding 95% CIs were estimated using the BrookmeyerCrowley method.

An independent data monitoring committee monitored the trial. The interim analyses were planned. Statistical analyses were conducted using SPSS (version 26.0), R (version 4.0.2). The trial is registered with the Chinese Clinical Trial Registry (number: ChiCTR1900023710).

## Reporting summary

Further information on research design is available in the Nature Portfolio Reporting Summary linked to this article.

## Data availability

All requests for data will be reviewed by the clinical site Sun Yat-sen University Cancer Centre and the study sponsor, Shanghai Junshi Biosciences Co., to verify if the request is subject to any intellectual property or confidentiality obligations. A proposal with detailed description of study objectives and statistical analysis plan will be needed for evaluation of the request. Additional materials might also be required during the process of evaluation. Data are available to request 12 months after the publication of this article. Requests for access to the de-identified participant data from this study can be submitted via email to lvxing@sysucc.org.cn with detailed proposal for approval. Please allow one month for response to requests. A signed data access agreement with the sponsor is required before accessing the shared data. The study protocol is available as Supplementary Note in the Supplementary Information file. The remaining data are available within the Article, Supplementary Information or Source Data file. Source data are provided with this paper.

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

## Acknowledgements

This study was supported by grants from the National Natural Science Foundation of China (No. 81872375, No. 82172863), the Natural Science Foundation of Guangdong Province (No. 2021A1515010118, No. 2019B110233004). The sponsor (Shanghai Junshi Biosciences Co.) pro-vided toripalimab free of charge for this study, and had no role in study design, conduct, data collection, data analysis or the writing of this report. We thank our patients and their families for their willingness to participate in this trial. The paper has been edited by Elsevier Language Editing Services.

## Author contributions

X.L. and X.G. were the principal investigators and participated in trial design, study management, data and toxicity review, review of the report, supervision of the study, and final approval of the report. Xu.C., C.X.L., J.Y.Z. and Xi.C. contributed to the writing of the protocol, recruitment and treatment of the patients, data and trial management, data analysis and interpretation, and writing of manuscript. H.Y.H., Y.Y.H, and Z.J.Z. participated in the recruitment and treatment of the patients, data and trial management, and report preparation. Xu.C., C.X.L., H.Y.H., and Z.C.L. were responsible for statistical analysis and interpretation as well as data review. L.R.K., L.J.H., W.X.X., L.Q.T., S.S.G. and H.L. contributed to patient accrual, toxicity review, and review of the completed report. All authors have reviewed and approved the final draft. All authors had full access to all the data in the study and the corresponding author had final responsibility for the decision to submit for publication.

## Competing interests

The authors declare no competing interests.

## Additional information

Xun Cao [1,2,3,6], Hao-Yang Huang[1,3,6], Chi-Xiong Liang[1,3,6], Zhuo-Chen Lin[4], Jia-Yu Zhou[1,3], Xi Chen[1,3], Ying-Ying Huang[3,5], Ze-Jiang Zhan[1,3], Liang-Ru Ke[3,5], Lu-Jun Han[3,5], Wei-Xiong Xia[1,3], Lin-Quan Tang[1,3], Shan-Shan Guo[1,3], Hu Liang[1,3], Xiang Guo[1,3,7] & Xing Lv [1,3,7] ✉

[1]Department of Nasopharyngeal Carcinoma, Sun Yat-sen University Cancer Centre, Guangzhou, China. [2]Department of Critical Care Medicine, Sun Yat-sen University Cancer Centre, Guangzhou, China. [3]State Key Laboratory of Oncology in South China/Collaborative Innovation Centre for Cancer Medicine/ Guangdong Key Laboratory of Nasopharyngeal Carcinoma Diagnosis and Therapy/Guangdong Provincial Clinical Research Centre for Cancer, Sun Yat-sen University Cancer Centre, Guangzhou, China. [4]Department of Medical Records, The First Affiliated Hospital of Sun Yat-sen University, Guangzhou, China. [5]Department of Medical Imaging, Sun Yat-sen University Cancer Centre, Guangzhou, China. [6]These authors contributed equally: Xun Cao, Hao-Yang Huang, Chi-Xiong Liang. [7]These authors jointly supervised this work: Xiang Guo, Xing Lv. ✉e-mail: lvxing@sysucc.org.cn

