## [Peer Review File · Nature Communications]

Toripalimab plus capecitabine in the treatment of patients with residual nasopharyngeal carcinoma: a single-arm phase 2 trialREVIEWER COMMENTS

Reviewer #1 (Remarks to the Author): with expertise in nasopharyngeal carcinoma

Comments to authors

1. Consider putting actual regimen in abstract.

2. Consider review for English language improvement. For example, line 31:

“More than 70% patient with nasopharyngeal carcinoma are diagnosed with locoregionally advanced disease at presentation, and in this subset of patients with an unfavorable prognosis”

Is very awkward. There are similar sentences throughout, but I will not list all of them here.

The first two sentences of the discussion are other examples.

3. line 40: I am not so sure that reirradiation is widely accepted as the most common approach to persistent LR disease, and only a very very small fraction of patients with disease persistence are eligible for surgical salvage. That should be stressed.

4. A major challenge with interpretation of this study is to disambiguate the contribution of each of the interventions. Authors cite references of a 20% RR for toripalimab in r/m NPC and 37-48% for capecitabine as single agents, and in the case of the latter this was seen in SECOND LINE R/M disease. So with mere additivity one might expect a RR of the combination to be > 60% in the R/M setting.

5. What was prior systemic drug exposure? This should be in the manuscript, not just appendix. It appears that 71% had prior 5-FU exposure. Was the impact of prior 5-FU exposure evaluated? There is no quantitative evaluation Of this, but it may not be possible with 23 patients.

6. The requirement for only radiographic evidence of persistence at 12-16 weeks post radiation is somewhat problematic, for it is not unusual for there to be, by conventional radiological imaging, residual abnormalities. In how many patients was there definitive histological evidence of persistent viable cancer or PET CT evidence c/w persistent biologically active disease? It is not clear how many/ which patients had PET CTs, but it seems like many had neither baseline PET CT nor biopsy.

8. I see no data concerning results of blood EBV DNA levels.

9. Were there interval between end of xrt and start of rx data? Should be close to 12-16 weeks.

10. It is not clear from the report, but it is assumed that these patients all did not have evidence of metastasis at enrollment. That is, there should be clarification that these were patients with ONLY LR persistent disease per exclusion criterion 1. This should be highlighted in the title and abstract, as this is a very narrow patient population, and expected to be biologically different from those with progression to metastatic sites. It is also notable that baseline evaluation included only MRI of NO and chest CT, not PET CT.

11. Line 263. It is unclear why exclusion for salvage surgery and reirradiation candidates would enrich for tori plus cape sensitivity. If the authors have a biological hypothesis for this, they should amplify on this here.

12. line 268. This is redundant with the intro and can be removed.

13. line 311. Orally administered systemic chemotherapy is not inherently less toxic than IV treatment.

14 line 327. "less toxic" than what?

15. line 330, It seems that authors are suggesting that HFS may be ameliorated by GCSF. I know of no data to support that claim and none is offered.

16. Line 335 and beyond. See comment above on this issue as well. A major challenge of study interpretation are the issues outlined here. It is possible, even likely, that there were patients enrolled on this study using radiographical criteria alone, whose tumors were still measurable but not viable. While in this paragraph and later in the discussion there is considerable effort to justify no confirmation biopsy, the possibility of additional treatment of potentially cured patients is not directly addressed. There is further discussion of this in the vicinity of line 362, but since this is the first study I know of this kind, the possibility I discuss above remains unaddressed and somewhat unique to this trial.

17. Acknowledgement of inability to separate tori effect from cape effect is to be commended.

18. line 370. More specificity on "a study has been initiated" would be beneficial.

19. line 374. The authors do not specifically declare an opinion on how far these data should be interpreted or what next steps should be. "May represent the future therapeutic strategy" leaves one wondering.

20. Table 2. Using RECIST criteria to declare a complete response in a patient who has recently received radiation is a challenge, especially in patients who presented with T4 disease (43%), for it is very common that there is distortion of nasopharyngeal and

parapharyngeal tissue which confounds definition of "CR" How did the investigators deal with this ?

Reviewer #2 (Remarks to the Author): with expertise in biostatistics, clinical trial study design

Major Comment to the Authors:

1. The protocol states that withdrawal Criteria include "Patients who fail to follow the treatment plan" and "Patients who participate in other chemotherapy, surgery, or experimental drug therapy during the trial."

If noncompliant patients were willing to be followed then failing to do so is an error, leading to potential selection biases. The usual strategy is summarized as "Off treatment is not off study". Under intent-to-treat, all patients should be followed whether compliant or not. Please clarify the investigators' procedure in such instances.

Given that lines 218, 219 state "Overall, three of 23 patients (13.0%) 219 discontinued the study treatment due to adverse events ...", and that all 23 apparently were analyzed as shown in Figure 1, this may

Minor Comments to Authors:

A. Please correct grammatical errors in lines 31-33:

"More than 70% patient with nasopharyngeal carcinoma are diagnosed with locoregionally advanced disease at presentation, and in this subset of patients with an unfavourable prognosis. Over the past few decades, much efforts ..."

B. Line 166 states that "No interim analyses were planned.", but of course an interim analysis was conducted in order to determine to proceed to the second stage of the Simon design.

C. Lines 130-131 state "Then, the patients were followed up every 3 months for the first 3 years, every 6 months for 4-5 years, 131 and annually thereafter. The details of follow-up are specified in the protocol." Is this relevant, given that the longest followup was 18 and, for the majority of patients, less than 12 months?

D. In the Discussion, Lines 263,4 you state "Therefore, this exclusion enriched the trial population for patients who may benefit from our study treatment. " I don't see that - you only excluded subjects who were unable to receive other treatments, which doesn't mean that they are more likely to benefit from yours. I think that you are being too modest here.

E. Lines 249-250 state that "This is the first clinical trial that focused on the management of persistent or residual nasopharyngeal 250 carcinoma." I have a hard time believing this to be true, especially since clinicaltrials.gov lists several completed studies.

Reviewer #3 (Remarks to the Author): with expertise in nasopharyngeal carcinoma

Response assessment

Is the persistent or recurrent disease truly measurable by RECIST?

Procedures (P6):

"For lesions that were not accessible, the clinical diagnosis was also accepted based on the presence of at least two classic radiological features (with or without clinical symptoms) on radiological imaging (details are shown in the protocol)."

However, I cannot locate the relevant radiological diagnostic criteria in the attached a

protocol (23 pages).

It is well known that tumour lesions situated in a previously irradiated area are usually not considered measurable unless there has been demonstrated progression in the lesion. Study protocols should detail the conditions under which such lesions would be considered measurable. As the location of persistent or recurrent lesions in this study were all within previously irradiated area (skull base or neck), this radiological inclusion criteria is critical for determining whether the lesion is measurable or not by RECIST criteria. Such information should be specifically included in the manuscript (or an appendix).

Procedures (P7):

Biochemical test, urine and faecal test...

Out of interest, which faecal test is relevant for this study protocol?

Efficacy:

Post-radiotherapy plasma EBV DNA load was demonstrated to stratify patients into distinct prognostic groups (Ann Oncol. 2020 Jun;31(6):769-779). Furthermore, the clearance of post-radiotherapy residual plasma EBV DNA strongly correlate with subsequent clinical course and disease status (Clin Cancer Res. 2021 May 15;27(10):2827-2836). It would be most interesting to present the dynamic changes of plasma EBV DNA load collected in this patient cohort treated with a combination of immune check point inhibitor and chemotherapy.

There were two disease progression documented for the 23 patients and all were loco-regional recurrence. It appeared that these two patients had complete response as their best tumor response. It would be interesting to know whether the site of progression was at previous target lesion or new sites recurrences. Is that any progression at distant sites?

Toxicity:

It was stated that the most common grade 1-2 immune related adverse events included hypothyroidism, fatigue, leukopenia, lymphopenia, anaemia. It is easy to understand hypothyroidism and fatigue would be most likely attributed to immune-related adverse events. However, leucopenia, lymphopenia and anemia were more commonly attributed to chemotherapy (capecitabine). Persistent lymphopenia for more than 6 months is a well-

documented phenomena after head and neck radiotherapy. How to ascertain the attribution as immune-related adverse events under these circumstances? Was the grade 3 myocardial infarction most likely related to capecitabine?

Authors' response

Reviewer #1

Thank you for your encouragement and insightful comments.

Reviewer's comments	Authors' responses or change made	Page number
1. Consider putting actual regimen in abstract.	Many thanks for your suggestion. We have included the actual regimen in the "Abstract" section.	Lines 13-14 on page 1
2. Consider review for English language improvement. For example, line 31: "More than 70% patient with nasopharyngeal carcinoma are diagnosed with locoregionally advanced disease at presentation, and in this subset of patients with an unfavorable prognosis". Is very awkward. There are similar sentences throughout, but I will not list all of them here. The first two sentences of the discussion are other examples.	Many thanks for your helpful suggestion. We have reworded the sentences in the "Introduction" and "Discussion" sections. According to the Reviewer's suggestion, the paper has been sent to Nature Research Editing Service for re-editing (No. AB79-4356-4274-C3FF-656P).	Lines 26-28 on page 2, and lines 231-233 on page 11
3. Line 40: I am not so sure that reirradiation is widely accepted as the most common approach to persistent LR disease, and only a very very small fraction of patients with disease persistence are eligible for surgical salvage. That should be stressed.	The nasopharynx is surrounded by bony structures and complex adjacency of the nasopharynx. Intensity-modulated radiotherapy (IMRT) is able to precisely deliver radiation to the target tissue while sparing the surrounding tissue. ¹⁻³ Thus, repeat irradiation has been the primary option in the IMRT era. ¹ It is acknowledged that surgery is the primary treatment approach for patients with isolated regional lymph nodes ⁴ or resectable persistent or residual lesions. ¹ These resectable areas are defined according to the 8 th edition of the International Union Against Cancer and American Joint Committee on Cancer staging system for nasopharyngeal carcinoma. ⁵⁻⁷	
4. A major challenge with interpretation of this study is to disambiguate the contribution of each of the interventions. Authors cite references of a 20% RR for	Thank you for your insightful comment. Based on the literature available at the time of this study planned, the highest objective response rate of capecitabine was 47.8% in recurrent nasopharyngeal carcinoma. ⁸ The objective response rate of immune	Lines 333-334 on page 16

toripalimab in r/m NPC and 37-48% for capecitabine as single agents, and in the case of the latter this was seen in SECOND LINE R/M disease. So, with mere additivity one might expect a RR of the combination to be > 60% in the R/M setting.

checkpoint inhibitors ranged from 20.5% to 34%.⁹⁻¹² In Fang and colleagues' study, 90.9% of patients had an objective response to chemotherapy and immune checkpoint inhibitor combined treatment.¹¹ We considered that the different mechanisms of the two drugs might induce synergistic anti-tumour effects.¹³ Therefore, the objective response rate of our combination regimen (toripalimab plus capecitabine) was initially expected to be 80%. Furthermore, we acknowledge that our study was a single-arm, phase 2 trial lacking a comparison arm to characterize the effect of each trial drug vs a dual regimen. This issue has been described in the limitation paragraph of the "Discussion" section.

5. What was prior systemic drug exposure? This should be in the manuscript, not just appendix. It appears that 71% had prior 5-FU exposure. Was the impact of prior 5-FU exposure evaluated? There is no quantitative evaluation. Of this, but it may not be possible with 23 patients.

Many thanks for this comment. The prior systemic drug exposure should be in the manuscript. In accordance with the Reviewer's comment, we have moved these Supplementary Tables to the manuscript. In our study, 21 (91.3%) of 23 patients received induction chemotherapy. The Reviewer is correct that the impact of fluorouracil exposure should be evaluated. In reply to this comment, these data have been provided to help the reader to better understand our results.

Line 177 on page 9, Tables 2 and 3, Lines 186-188 on page 9, and Supplementary Table 2

The results are presented as follows:

Table. Tumour response to study treatment according to previous exposure to fluorouracil

	Efficacy population (n=23)			
	Previous exposure to fluorouracil (n=15)		Non- previous exposure to fluorouracil (n=8)	
Complete response	7	(46.7%)	6	(75%)
Partial response	7	(46.7%)	2	(25%)
Stable disease	1	(6.67%)	0	
Objective response rate	14	(93.3%; 68.1-99.8)	8	(100%; 100-100)
Complete response rate	7	(46.7%; 21.3-73.4)	6	(75%; 34.9-96.8)
Disease control rate	15	(100%; 100-100)	8	(100%; 100-100)

Data are n (%) or n (%; 95% CI). Percentages (%) might not total 100% because of rounding. 95% CI=95% confidence interval.

6. The requirement for only radiographic evidence of persistence at 12-16 weeks post radiation is somewhat problematic, for it is not unusual for there to be, by conventional radiological imaging, residual abnormalities. In how many patients was there definitive histological evidence of persistent viable cancer or PET-CT evidence c/w persistent biologically active disease? It is not clear how many/which patients had PET-CTs, but it seems like many had neither baseline PET-CT nor biopsy.

Many thanks for your insightful comment. The Reviewer is correct that both histological evidence and radiological evidence of persistent or residual diseases are important. However, although biopsy or fine needle aspiration is well recognized as the gold standard for the diagnosis of residual tumours, the routine use of these techniques is not preferred because their invasive property.¹⁴ In addition, even for residual lesions in the nasopharynx, the chance that a single biopsy will miss residual tumours is 26.4%.¹⁵ In He and colleagues' study, only 11.5% (27/236) of patients had lesions that could be accessed by biopsy.¹⁶ Thus, for lesions that were not accessible by biopsy or fine needle aspiration, the radiographic diagnoses were accepted, which has also been reported by some other studies.^{14,16-18}

Lines 175-176 on page 8, Supplementary Table 1

MRI is currently the most commonly used imaging modalities for persistent or residual nasopharyngeal carcinoma diagnosis.^{15,19} Here, in response to the Reviewer's comment, we have expanded the descriptions of histological and radiographic evidence of persistent or residual disease in our revised manuscript.

Table. Diagnostic evidence of persistent or residual nasopharyngeal carcinoma

	All patients (n=23)	
	n	(%)
Histological diagnosis *	4	(17.4%)
Radiographic diagnosis	19	(82.6%)
By MRI	11	(47.8%)
By MRI and ¹⁸ F-fluorodeoxyglucose PET-CT	8	(34.8%)

* All patients underwent MRI. MRI= magnetic resonance imaging.

7. I see no data concerning results of blood EBV DNA levels.

Many thanks for this comment. In our trial, all patients were negative for Epstein-Barr virus DNA (EBV-DNA) after 6 cycles of toripalimab plus capecitabine combination

Lines 174-175 on page 8, Lines

treatment (all patients achieved negative EBV-DNA by 3 cycles). The results of the EBV-DNA analysis have been provided in our revised manuscript.

189-190 on page 9, Table 1, Supplementary Table 3 and Supplementary Figure 1

Table. Dynamic change of Epstein-Barr virus DNA during our study treatment

	All patients (n=23)		
	Baseline (n [%])	At the end of 3 cycles of study treatment (n [%])	After completion of 6 cycles of study treatment (n [%])
Detectable	2 (8.7%)	0	0
Undetectable	21 (91.3%)	23 (100%)	23 (100%)

Figure. Longitudinal trajectory Epstein-Barr virus DNA trend over our study treatment

8. Were there interval between end of xrt and start of rx data? Should be close to 12-16 weeks. Thank you for this query. The interval between the final radiation dose (xrt) and the start of receiving data (rx) was approximately 12-16 weeks.
9. It is not clear from the report, but it is assumed that these patients all did not have evidence of metastasis at enrollment. That is, there should be clarification that these were patients with ONLY LR persistent disease per exclusion criterion 1. This should be highlighted in the title and abstract, as this is a very narrow patient population, and expected to be biologically different from those with progression to metastatic sites. It is also notable that baseline evaluation included only MRI of NO and chest CT, not PET CT. **Table. Radiographic evaluation of persistent or residual nasopharyngeal carcinoma**
- | | | All patients (n=23) | |
|---|--|----------------------------|------------|
| | | n | (%) |
| MRI of nasopharynx and neck | | 15 | (65.2%) |
| MRI and ¹⁸ F-fluorodeoxyglucose PET-CT | | 8 | (34.8%) |
| CT scan of chest | | 23 | (100%) |
10. Line 263. It is unclear why exclusion for salvage surgery and re-irradiation candidates would enrich for tori plus cape sensitivity. If the authors have a biological hypothesis for this, they should amplify on this here. Thank you for your professional comment. As always, we aimed to identify an effective treatment regimen for patients with locally persistent or residual nasopharyngeal carcinoma. The treatments for persistent or residual disease include re-irradiation, surgery, and chemotherapy. Re-irradiation has been the primary option in the intensity-modulated radiotherapy (IMRT) era;¹ surgery is the primary treatment approach for patients with isolated regional lymph nodes⁴ or resectable persistent or residual lesions. Some patients will benefit from these local treatments.¹ However, toxic effects and complications remain pertinent issues. Therefore, in the present trial, we enrolled patients with locally persistent or residual disease that is neither resectable nor suitable for re-irradiation. We think that these patients will benefit from the combination treatment in our study
11. Line 268. This is redundant with the intro and can be removed. Many thanks for your suggestion. The sentence has been removed.
- Line 1, line 5, line 10, and line 19 on page 1
- Line 242 in page 11, and lines 243-247 in page 12

12.	Line 311. Orally administered systemic chemotherapy is not inherently less toxic than IV treatment.	We thank you for your thoughtful reminder. The sentence has been reworded.	Lines 292-293 on page 14
13.	Line 327. “less toxic” than what?	Many thanks for your suggestion. We have reworded the sentence in the “Discussion” section	Line 306 on page 14
14.	Line 330, It seems that authors are suggesting that HFS may be ameliorated by GCSF. I know of no data to support that claim and none is offered.	Many thanks for this correction. We have corrected the clerical errors in this sentence.	Lines 307-308 on page 14, and lines 309-310 on page 15
15.	Line 335 and beyond. See comment above on this issue as well. A major challenge of study interpretation are the issues outlined here. It is possible, even likely, that there were patients enrolled on this study using radiographical criteria alone, whose tumors were still measurable but not viable. While in this paragraph and later in the discussion there is considerable effort to justify no confirmation biopsy, the possibility of additional treatment of potentially cured patients is not directly addressed. There is further discussion of this in the vicinity of line 362, but since this is the first study I know of this kind, the possibility I discuss above remains unaddressed and somewhat unique to this trial.	We appreciate your professional suggestion on this matter. In our trial, the residual tumour was confirmed by biopsy or fine needle aspiration; radiographic diagnosis was also accepted for lesions that were not accessible.^{14,16-18} Given the results of previous studies, we thought that imaging modalities (eg, MRI and ¹⁸F-fluorodeoxyglucose [¹⁸F-FDG] PET-CT) could be suitable for the evaluation and diagnosis of persistent or residual tumours.^{20,21} Combined MRI and ¹⁸F-FDG PET-CT is more accurate for persistent or residual nasopharyngeal carcinoma diagnosis.^{1,20,22} This is the first clinical trial focusing on the management of persistent or residual nasopharyngeal carcinoma. We acknowledged the possibility that additional study treatment might have been administered to some patients who potentially could have been cured with primary treatments alone. Nevertheless, by combining MRI and ¹⁸F-FDG PET-CT,²² we have minimized these possibilities. In addition, in 2021, Chen and colleagues reported that the addition of adjuvant capecitabine to chemoradiotherapy significantly improved 3-year failure-free survival in patients with locoregionally advanced nasopharyngeal carcinoma after receiving standard-of-care treatment.²³ Regarding anti-programmed cell death-1 (PD-1) monoclonal antibodies, one phase 3 trial (NCT03427827) investigating adjuvant camrelizumab in locoregionally advanced nasopharyngeal carcinoma will test the value of adding anti-PD-1 therapy to standard	

treatment in the curative setting. The results of these studies will provide more evidence. Even if the effects are unique, our study's results have shown promising anti-tumour activity and a predictable safety profile.

- | | | |
|--|---|---|
| 16. Acknowledgement of inability to separate tori effect from cape effect is to be commended. | We appreciate your helpful comment. We acknowledge that it is difficult to distinguish the adverse effects due to toripalimab from those due capecitabine. This point has been described in the limitations paragraph of the "Discussion" section in our revised manuscript. | Lines 347-348 on page 16 |
| 17. Line 370. More specificity on "a study has been initiated" would be beneficial. | Many thanks for your insightful suggestion. In our trial, the absence of viable tumour cells in the biopsy or fine needle aspiration specimens of patients hindered prognostic analysis. The matched blood samples collected may provide more valuable information. A study to investigate tumour mutational burden and genetic biomarkers by whole-exome sequencing (eg, microsatellite stability status, nucleotide variants, short and long insertions and deletions [INDELs], copy number variants, and gene rearrangement and fusions) in blood samples as potential predictors of efficacy has been initiated. We have specified this section in our revised manuscript | Lines 351-352 on page 16, and line 353 on page 17 |
| 18. Line 374. The authors do not specifically declare an opinion on how far these data should be interpreted or what next steps should be. "May represent the future therapeutic strategy" leaves one wondering. | We greatly appreciate your helpful reminder. Our study suggested that the combination of toripalimab and capecitabine had promising efficacy with a manageable safety profile for treating patients with locally persistent or residual nasopharyngeal carcinoma. However, the median progression-free survival and the median duration of response were not reached within the median follow-up of 11.1 months. As specified in the protocol, progressive disease (including local and distant failure) will be documented further. All enrolled patients will be followed up until death. Then, the progression-free survival and duration of response may further confirm that toripalimab plus capecitabine provides more benefit to patients with locally persistent or residual diseases. These findings will demonstrate that toripalimab plus capecitabine as combination treatment could be considered a treatment option for patients with locally persistent or residual | |

19. Table 2. Using RECIST criteria to declare a complete response in a patient who has recently received radiation is a challenge, especially in patients who presented with T4 disease (43%), for it is very common that there is distortion of nasopharyngeal and parapharyngeal tissue which confounds definition of “CR” How did the investigators deal with this?	nasopharyngeal carcinoma that is neither resectable nor suitable for re-irradiation. We appreciate your professional suggestion on this matter. In our trial, treatment responses were assessed by the independent review team according to Response Evaluation Criteria in Solid Tumours (RECIST; version 1.1).^{24,25} Given the distortions of nasopharyngeal and parapharyngeal tissue and complex bony structures, assessment of tumour response can be challenging. However, in our study, if persistent or residual disease involved the destruction of bone, the bone response criteria described by The University of Texas MD Anderson Cancer Centre were used.^{26,27} Additionally, ¹⁸F-fluorodeoxyglucose (¹⁸F-FDG) PET-CT represents a function-based imaging criterion. The tumour margins that are necessary for reproducible anatomic measurements are of lesser importance in functional imaging.²⁷ Therefore, in the evaluation of tumour response, ¹⁸F-FDG PET-CT is also important to address the distortion of nasopharyngeal and parapharyngeal tissue. The fixed nasopharynx is located in the skull bone and therefore does not exhibit a profound change in tumour size despite the effectiveness of our study treatment.²⁸ By reflecting changes in tumour metabolism, ¹⁸F-FDG PET-CT provides a method by which tumour response can be measured in the absence of marked anatomic changes.²⁹ Therefore, in our trial, the combined use of MRI and ¹⁸F-FDG PET-CT could satisfactorily evaluate the tumour response with high accuracy.¹
---	--

Reviewer #2

Thank you for your encouragement and professional comments and suggestions.

Reviewer’s comments	Authors’ responses or change made	Page number
Major Comment:		
1. The protocol states that withdrawal Criteria include “Patients who fail to follow the treatment plan” and	We are very grateful for your insightful comment. The Reviewer is correct that “off treatment is not off study”. In clinical trials, the intention-to-treat population include all	Lines 496-499 on page 19 (on the

“Patients who participate in other chemotherapy, surgery, or experimental drug therapy during the trial.” If noncompliant patients were willing to be followed then failing to do so is an error, leading to potential selection biases. The usual strategy is summarized as “Off treatment is not off study”. Under intent-to-treat, all patients should be followed whether compliant or not. Please clarify the investigators’ procedure in such instances.

Given that lines 218, 219 state “Overall, three of 23 patients (13.0%) 219 discontinued the study treatment due to adverse events ...”, and that all 23 apparently were analyzed as shown in Figure 1, this may

assigned patients, and they should be followed whether they are compliant or not. In our study, efficacy and safety analyses were conducted for all assigned patients who received at least one dose of the study medications (efficacy population and safety population). These definitions of “efficacy population” and “safety population” have been used in other trials.^{11,30,31} Including the 3 patients who discontinued the study treatment due to adverse events and withdrawal of consent, all of the assigned patients are following up. The data can be found in the Kaplan–Meier curves of progression-free survival and duration of response. In response to the Reviewer’s comment, we have corrected the clerical errors in the manuscript. Many thanks for your helpful comment.

protocol);
Supplementary
Figure 2; lines
151-154 on page
7

Minor Comments:

1. Please correct grammatical errors in lines 31-33: “More than 70% patient with nasopharyngeal carcinoma are diagnosed with locoregionally advanced disease at presentation, and in this subset of patients with an unfavourable prognosis. Over the past few decades, much efforts ...”

Many thanks for this correction. We have corrected and reworded the sentence. In addition, the paper has been sent to Nature Research Editing Service for re-editing (No. AB79-4356-4274-C3FF-656P).

Lines 26-28 on
page 2

2. Line 166 states that “No interim analyses were planned.”, but of course an interim analysis was conducted in order to determine to proceed to the second stage of the Simon design.

We appreciate your expert suggestion. An interim analysis would be performed after the completion of stage one in the Simon design. We have corrected this error in statistical analysis paragraph of “Methods” section.

Line 162 on page
8

3. Lines 130-131 state “Then, the patients were followed up every 3 months for the first 3 years, every 6 months

Thank you for your comment. It is important to report the follow-up information to help the readers better understand the survival outcomes. At the cut-off date of December 10,

Line 127-129 on
page 9

for 4-5 years, and annually thereafter. The details of follow-up are specified in the protocol.” Is this relevant, given that the longest follow-up was 18 and, for the majority of patients, less than 12 months?

2021, the median follow-up duration was 11.1 months. As specified in the protocol, progressive disease will be documented further. All enrolled patients will be followed up until death. Then, the progression-free survival and duration of response may further confirm that toripalimab plus capecitabine may provides more benefit to patients with locally persistent or residual diseases. According to the Reviewer’s comment, we have expanded the follow-up information.

Table. Follow-up information

Follow-up time (month)	All patients (n=23)	
	n	(%)
IQR	9.2-13.6	
Range	7.2-17.8	
≥12 months	10	(43.5%)
<12 months	13	(56.5%)

IQR=interquartile range.

4. In the Discussion, Lines 263, 4 you state “Therefore, this exclusion enriched the trial population for patients who may benefit from our study treatment.” I don’t see that - you only excluded subjects who were unable to receive other treatments, which doesn’t mean that they are more likely to benefit from yours. I think that you are being too modest here.

Thank you for this comment. As always, we aim to identify an effective treatment regimen for patients with locally persistent or residual nasopharyngeal carcinoma. The treatments for residual or persistent disease include radiotherapy, surgery, and chemotherapy. Re-irradiation has been the primary option in the intensity-modulated radiotherapy (IMRT) era;¹ surgery is the primary treatment approach for patients with isolated regional lymph nodes⁴ and resectable lesions. Some patients may benefit from these local treatments.¹ However, toxic effects and complications remain pertinent issues. In this trial, we focused on patients with locally persistent or residual disease that is neither resectable nor suitable for re-irradiation. Therefore, we think that this trial population will benefit from the combination treatment in our study.

Line 242 in page 11, and lines 243-247 in page 12

5. Lines 249-250 state that “This is the first clinical trial that focused on the management of persistent or residual

Many thanks for your helpful comment. We acknowledge that several completed studies are listed at clinicaltrials.gov but have not yet reported results. We searched PubMed for

nasopharyngeal carcinoma.” I have a hard time believing this to be true, especially since clinicaltrials.gov lists several completed studies.

published clinical trials from database inception until December 10, 2021 (the data cut-off date) using the search terms (“nasopharyngeal carcinoma” OR “nasopharyngeal cancer”) AND (“persistent” OR “residual”) AND (“programmed cell death-1 [PD-1]” OR “programmed cell death-ligand 1 [PD-L1]” OR “immune checkpoint inhibitor” OR “immunotherapy”), with no language restrictions. No matching clinical trials were identified. To our knowledge, there is no published trial focusing on persistent or residual nasopharyngeal carcinoma and no evidence of the efficacy and safety of PD-1 inhibitors combined with oral chemotherapy as a treatment for this disease. To confirm this, using the same search terms, we searched PubMed for published clinical trials from database inception until March 19, 2022, and found no matching articles. Many thanks for your reminder.

Reviewer #3

Thank you for your encouragement and professional comments and suggestions.

Reviewer’s comments	Authors’ responses or change made	Page number
1. Response assessment Is the persistent or recurrent disease truly measurable by RECIST?	We appreciate your professional query on this matter. In our trial, responses to treatment were assessed by the independent review team using Response Evaluation Criteria in Solid Tumours (RECIST; version 1.1).^{24,25} Given the complex adjacency of the nasopharynx and surrounding complex bony structures, the assessment of tumour response can be challenging. However, in our study, if the persistent or residual disease involved the destruction of bone, the bone response criteria established by The University of Texas MD Anderson Cancer Centre were used.^{26,27} Additionally, ¹⁸F-fluorodeoxyglucose (¹⁸F-FDG) PET-CT is a function-based imaging modality. The tumour margins that are necessary for reproducible anatomic measurements are of lesser importance in functional imaging.²⁷ Therefore, in the evaluation of tumour response, ¹⁸F-FDG PET-CT is important for identifying	

persistent or residual nasopharyngeal carcinoma. Persistent or residual lesions that are located in the fixed complex nasopharynx and may not exhibit a profound tumour volume response despite the effectiveness of our study treatment.²⁸ By reflecting changes in tumour metabolism, ¹⁸F-FDG PET-CT provides a method by which tumour response can be measured in the absence of marked anatomic changes.²⁹ Therefore, in our trial, the combination of MRI and ¹⁸F-FDG PET-CT could be used to evaluate tumour response with satisfactory accuracy.

2. Procedures (P6):

“For lesions that were not accessible, the clinical diagnosis was also accepted based on the presence of at least two classic radiological features (with or without clinical symptoms) on radiological imaging (details are shown in the protocol).”

However, I cannot locate the relevant radiological diagnostic criteria in the attached a protocol (23 pages). It is well known that tumour lesions situated in a previously irradiated area are usually not considered measurable unless there has been demonstrated progression in the lesion. Study protocols should detail the conditions under which such lesions would be considered measurable. As the location of persistent or recurrent lesions in this study were all within previously irradiated area (skull base or neck), this radiological inclusion criteria is critical for determining whether the lesion is measurable or not by RECIST criteria. Such information should be specifically included in the

We appreciate your professional suggestion on this matter. It is important to specify the radiological diagnostic criteria, which are critical for determining whether the lesion is measurable by RECIST. We have provided the radiological diagnostic criteria in the Supplementary Appendix of our revised manuscript.

lines 101-102 in page 5, and the Supplementary Appendix (page 6)

manuscript (or an appendix).

3. Procedures (P7):
Biochemical test, urine and faecal test...
Out of interest, which faecal test is relevant for this study protocol?

In our trial, the faecal test was routinely performed before study treatment.

4. Efficacy:
Post-radiotherapy plasma EBV DNA load was demonstrated to stratify patients into distinct prognostic groups (Ann Oncol. 2020 Jun;31(6):769-779). Furthermore, the clearance of post-radiotherapy residual plasma EBV DNA strongly correlate with subsequent clinical course and disease status (Clin Cancer Res. 2021 May 15;27(10):2827-2836). It would be most interesting to present the dynamic changes of plasma EBV DNA load collected in this patient cohort treated with a combination of immune check point inhibitor and chemotherapy.

Thank you for your insightful comment. The Reviewer is correct that the detection of Epstein–Barr virus DNA (EBV-DNA) is important for patient stratification and disease surveillance.^{1,32-35} Several studies have reported that the incidence of detectable post-treatment EBV-DNA ranged from 6.8% to 7.6%.^{36,37} These incidences were consistent with our results. As the Reviewer indicates, post-treatment EBV-DNA is an important prognostic factor in nasopharyngeal carcinoma.^{32,33} In 2018, Chan and colleagues reported that post-treatment EBV-DNA (detectable *vs* undetectable) was inconsistent with post-treatment ¹⁸F-fluorodeoxyglucose PET-CT (positive *vs* negative).³⁸ Therefore, the associations between post-treatment EBV-DNA and tumour residue remain to be resolved. In our trial, all patients were negative for EBV-DNA after 6 cycles of toripalimab plus capecitabine combination treatment (all patients achieved negative EBV-DNA by 3 cycles). In accordance with the Reviewer’s suggestion, the dynamic changes and longitudinal trajectory of EBV-DNA in the trial population have been provided.

Lines 189-190 on page 9, Table 1, Supplementary Table 3, and Supplementary Figure 1

Table. Dynamic change of Epstein-Barr virus DNA during our study treatment

	Baseline (n [%])	At the end of three cycles of scheduled treatment (n [%])	After completion of six cycles of scheduled treatment (n [%])
Detectable	2 (8.7%)	0	0
Undetectable	21 (91.3%)	23 (100%)	23 (100%)

Figure. Longitudinal trajectory of Epstein-Barr virus DNA trend over our study

treatment

5. There were two disease progression documented for the 23 patients and all were loco-regional recurrence. It appeared that these two patients had complete response as their best tumor response. It would be interesting to know whether the site of progression was at previous target lesion or new sites recurrences. Is that any progression at distant sites?

We thank the Reviewer for this valuable suggestion. Two disease progression events were recorded, both of which involved locoregional recurrence at the previous target lesion. No distant metastasis was recorded. We have added relevant data in the “Results” section.

Lines 201-203 on page 10

6. Toxicity:
It was stated that the most common grade 1-2 immune related adverse events included hypothyroidism, fatigue,

Many thanks for your comment. We acknowledge that persistent lymphopenia for more than 6 months is a well-documented phenomenon after head and neck radiotherapy. In our trial, immune-related adverse events were evaluated by the investigators, and three

leukopenia, lymphopenia, anaemia. It is easy to understand hypothyroidism and fatigue would be most likely attributed to immune-related adverse events. However, leucopenia, lymphopenia and anaemia were more commonly attributed to chemotherapy (capecitabine). Persistent lymphopenia for more than 6 months is a well-documented phenomena after head and neck radiotherapy. How to ascertain the attribution as immune-related adverse events under these circumstances? Was the grade 3 myocardial infarction most likely related to capecitabine?

patients were reported to have immune-related lymphopenia. All of these patients have other commonly immune-related adverse events (eg, hypothyroidism, fatigue, and rash) concomitantly. Furthermore, abnormal cytokine levels were also observed in these patients. Therefore, we could ascertain that these adverse events were attributable to the immunotherapy (toripalimab).

First, after reviewing the instructions of the administered drugs,³⁹⁻⁴¹ we found that the incidence of myocardial infarction related to capecitabine in completed clinical trials was approximately 5%.⁴¹ After consultation with our oncologists and cardiologists, we concluded that the grade 3 myocardial infarction might be related to the capecitabine.

References:

- 1 Chen, Y. P. et al. Nasopharyngeal carcinoma. *Lancet*. **394**, 64-80 (2019).
- 2 Stoker, S. D. et al. Current treatment options for local residual nasopharyngeal carcinoma. *Curr Treat Options Oncol*. **14**, 475-491 (2013).
- 3 Liu, F. et al. Fractionated stereotactic radiotherapy for 136 patients with locally residual nasopharyngeal carcinoma. *Radiat Oncol*. **8**, 157 (2013).
- 4 Liu, Y. P. et al. Surgery for isolated regional failure in nasopharyngeal carcinoma after radiation: Selective or comprehensive neck dissection. *Laryngoscope*. **129**, 387-395 (2019).
- 5 Liu, Y. P. et al. Endoscopic surgery compared with intensity-modulated radiotherapy in resectable locally recurrent nasopharyngeal carcinoma: a multicentre, open-label, randomised, controlled, phase 3 trial. *Lancet Oncol*. **22**, 381-390 (2021).
- 6 Chen, M. Y. et al. Endoscopic nasopharyngectomy for locally recurrent nasopharyngeal carcinoma. *Laryngoscope*. **119**, 516-522 (2009).
- 7 Amin, M. B. *AJCC cancer staging manual*. (American Joint Committee on Cancer, Springer, 2017).
- 8 Ciuleanu, E. et al. Capecitabine as salvage treatment in relapsed nasopharyngeal carcinoma: a phase II study. *J BUON*. **13**, 37-42 (2008).
- 9 Hsu, C. et al. Safety and Antitumor Activity of Pembrolizumab in Patients With Programmed Death-Ligand 1-Positive Nasopharyngeal Carcinoma: Results of the KEYNOTE-028 Study. *J Clin Oncol*. **35**, 4050-4056 (2017).
- 10 Ma, B. B. Y. et al. Antitumor Activity of Nivolumab in Recurrent and Metastatic Nasopharyngeal Carcinoma: An International, Multicenter Study of the Mayo Clinic

- Phase 2 Consortium (NCI-9742). *J Clin Oncol.* **36**, 1412-1418 (2018).
- 11 Fang, W. et al. Camrelizumab (SHR-1210) alone or in combination with gemcitabine plus cisplatin for nasopharyngeal carcinoma: results from two single-arm, phase 1 trials. *Lancet Oncol.* **19**, 1338-1350 (2018).
- 12 Xu, R. et al. Recombinant humanized anti-PD-1 monoclonal antibody (JS001) in patients with refractory/metastatic nasopharyngeal carcinoma: Preliminary results of an open-label phase II clinical study. *Ann Oncol.* **29**, viii409 (2018).
- 13 Bracci, L., Schiavoni, G., Sistigu, A. & Belardelli, F. Immune-based mechanisms of cytotoxic chemotherapy: implications for the design of novel and rationale-based combined treatments against cancer. *Cell Death Differ.* **21**, 15-25 (2014).
- 14 Lv, J. W. et al. Magnetic Resonance Imaging-Detected Tumor Residue after Intensity-Modulated Radiation Therapy and its Association with Post-Radiation Plasma Epstein-Barr Virus Deoxyribonucleic Acid in Nasopharyngeal Carcinoma. *J Cancer.* **8**, 861-869 (2017).
- 15 Kwong, D. L. et al. The time course of histologic remission after treatment of patients with nasopharyngeal carcinoma. *Cancer.* **85**, 1446-1453 (1999).
- 16 He, Y. et al. A retrospective study of the prognostic value of MRI-derived residual tumors at the end of intensity-modulated radiotherapy in 358 patients with locally-advanced nasopharyngeal carcinoma. *Radiat Oncol.* **10**, 89 (2015).
- 17 Lin, G. W., Wang, L. X., Ji, M. & Qian, H. Z. The use of MR imaging to detect residual versus recurrent nasopharyngeal carcinoma following treatment with radiation therapy. *Eur J Radiol.* **82**, 2240-2246 (2013).
- 18 Ng, S. H. et al. Comprehensive imaging of residual/ recurrent nasopharyngeal carcinoma using whole-body MRI at 3 T compared with FDG-PET-CT. *Eur Radiol.* **20**, 2229-2240 (2010).
- 19 Chen, Y. P. et al. Induction Chemotherapy plus Concurrent Chemoradiotherapy in Endemic Nasopharyngeal Carcinoma: Individual Patient Data Pooled Analysis of Four Randomized Trials. *Clin Cancer Res.* **24**, 1824-1833 (2018).
- 20 Comoretto, M. et al. Detection and restaging of residual and/or recurrent nasopharyngeal carcinoma after chemotherapy and radiation therapy: comparison of MR imaging and FDG PET/CT. *Radiology.* **249**, 203-211 (2008).
- 21 Yen, R. F. et al. 18-fluoro-2-deoxyglucose positron emission tomography in detecting residual/recurrent nasopharyngeal carcinomas and comparison with magnetic resonance imaging. *Cancer.* **98**, 283-287 (2003).
- 22 Wei, J., Pei, S. & Zhu, X. Comparison of 18F-FDG PET/CT, MRI and SPECT in the diagnosis of local residual/recurrent nasopharyngeal carcinoma: A meta-analysis. *Oral Oncol.* **52**, 11-17 (2016).
- 23 Chen, Y. P. et al. Metronomic capecitabine as adjuvant therapy in locoregionally advanced nasopharyngeal carcinoma: a multicentre, open-label, parallel-group, randomised, controlled, phase 3 trial. *Lancet.* **398**, 303-313 (2021).

- 24 Response Evaluation Criteria in Solid Tumours <https://recist.eortc.org> (2009).
- 25 Eisenhauer, E. A. et al. New response evaluation criteria in solid tumours: revised RECIST guideline (version 1.1). *Eur J Cancer*. **45**, 228-247 (2009).
- 26 Hamaoka, T., Madewell, J. E., Podoloff, D. A., Hortobagyi, G. N. & Ueno, N. T. Bone imaging in metastatic breast cancer. *J Clin Oncol*. **22**, 2942-2953 (2004).
- 27 Costelloe, C. M., Chuang, H. H., Madewell, J. E. & Ueno, N. T. Cancer Response Criteria and Bone Metastases: RECIST 1.1, MDA and PERCIST. *J Cancer*. **1**, 80-92 (2010).
- 28 Michaelis, L. C. & Ratain, M. J. Measuring response in a post-RECIST world: from black and white to shades of grey. *Nat Rev Cancer*. **6**, 409-414 (2006).
- 29 Van den Abbeele, A. D. The lessons of GIST--PET and PET/CT: a new paradigm for imaging. *Oncologist*. **13 Suppl 2**, 8-13 (2008).
- 30 Shi, Y. et al. Safety and activity of sintilimab in patients with relapsed or refractory classical Hodgkin lymphoma (ORIENT-1): a multicentre, single-arm, phase 2 trial. *Lancet Haematol*. **6**, e12-e19 (2019).
- 31 Cheng, H. et al. Camrelizumab plus apatinib in patients with high-risk chemorefractory or relapsed gestational trophoblastic neoplasia (CAP 01): a single-arm, open-label, phase 2 trial. *Lancet Oncol*. **22**, 1609-1617 (2021).
- 32 Hui, E. P. et al. Integrating postradiotherapy plasma Epstein-Barr virus DNA and TNM stage for risk stratification of nasopharyngeal carcinoma to adjuvant therapy. *Ann Oncol*. **31**, 769-779 (2020).
- 33 Hui, E. P. et al. Dynamic Changes of Post-Radiotherapy Plasma Epstein-Barr Virus DNA in a Randomized Trial of Adjuvant Chemotherapy Versus Observation in Nasopharyngeal Cancer. *Clin Cancer Res*. **27**, 2827-2836 (2021).
- 34 Guo, R. et al. Proposed modifications and incorporation of plasma Epstein-Barr virus DNA improve the TNM staging system for Epstein-Barr virus-related nasopharyngeal carcinoma. *Cancer*. **125**, 79-89 (2019).
- 35 Lee, V. H. et al. The addition of pretreatment plasma Epstein-Barr virus DNA into the eighth edition of nasopharyngeal cancer TNM stage classification. *Int J Cancer*. **144**, 1713-1722 (2019).
- 36 Chan, A. T. et al. Plasma Epstein-Barr virus DNA and residual disease after radiotherapy for undifferentiated nasopharyngeal carcinoma. *J Natl Cancer Inst*. **94**, 1614-1619 (2002).
- 37 Lv, J. et al. Liquid biopsy tracking during sequential chemo-radiotherapy identifies distinct prognostic phenotypes in nasopharyngeal carcinoma. *Nat Commun*. **10**, 3941 (2019).
- 38 Chan, A. T. C. et al. Analysis of Plasma Epstein-Barr Virus DNA in Nasopharyngeal Cancer After Chemoradiation to Identify High-Risk Patients for Adjuvant Chemotherapy: A Randomized Controlled Trial. *J Clin Oncol*, JCO2018777847 (2018).
- 39 National Medical Products Administration <https://www.nmpa.gov.cn>

- 40 Tuo Yi ® (Toripalimab Injection) Instruction <https://www.junshipharma.com> (2021).
- 41 XELODA ® (Capecitabine Tablets) Instruction <https://www.roche.com.cn> (2021).

REVIEWERS' COMMENTS:

Reviewer #3 (Remarks to the Author):

The authors had tried very hard to address most of the comments. There is remaining concern on the methodology to assess the true treatment efficacy in locally recurrent NPC.

For this analysis, a total of 23 subjects were recruited within 12 months from June 2020 to May 2021 with data cut-off date in Dec 2021. The median follow up was less than 12 months for the entire cohort. With such a short follow up, It will be premature to make a definitive conclusion on the true efficacy of this combination approach.

The location of the persistent or residual disease (Table 1) were at skull base (5/23, 21.7%) or skull base plus cervical lymph nodes (9/23, 39%), all within previously irradiated area.

First, It was well known that it was extremely difficult (if not impossible) by conventional imaging (MRI or CT scan, or even with PET-CT scan) to determine if any persistent or residual radiological abnormalities at skull base at 12-16 weeks post-RT to be post-RT inflammation/fibrosis vs genuine recurrence, unless there is unequivocal radiological progression on serial follow up images.

Secondly, for target lesion eligible for RECIST response criteria, tumor lesions situated in a previously irradiated area are usually not considered measurable unless there has been demonstrated progression in the lesion. As RECIST response rate was chosen as the primary endpoint of this trial, the inclusion of such RECIST non-measurable target lesions would raise uncertainty as to the validity of the exceptional high complete and partial response rate reported in this study. Progression free survival with adequate length of follow up would be a more clinically valid endpoint to assess the true efficacy of this combination for local recurrent diseases after prior chemoradiation.

It was noted that pathological diagnoses were not obtained from most enrolled subjects and persistent or residual NPC was mainly diagnosed by imaging in this study. This is a well-recognized limitation on the interpretation of the efficacy result in locally recurrent NPC

when the target lesions of interest were within previously irradiated area. It was well known that there is a high false positive rate for radiological abnormality in the skull base at the immediately post-RT period. The enrolment of subjects with false positive radiological abnormalities without histological confirmation or serial imaging to confirm true progression may have contributed to the falsely high complete response rate. Furthermore, nearly all patients (95.7%) had negative plasma EBV DNA load at enrolment, further supporting that the suspicion that many of these radiological abnormalities may be false positive lesions that often regress or remain stable on serial follow up images.

It should also be acknowledged that plasma EBV DNA may not be detectable in most patient with small local recurrence. For reference, the sensitivity of EBV DNA by PCR assay was only up to 42.3% for the detection of local recurrence in a recent study.

([https://www.annalsofoncology.org/article/S0923-7534\(22\)00753-0/fulltext](https://www.annalsofoncology.org/article/S0923-7534(22)00753-0/fulltext))

Despite all the above limitations, which were inherent in the interpretation of all studies in locally recurrent NPC, I still think that this study presented very interesting albeit preliminary findings in this difficult to treat situation where there is paucity of clinical trial results. Therefore, it would be important for the authors to highlight the above limitations in the interpretation of study results.

Other minor comments:

Title of Table 2, "front induction chemotherapy". Should it be "upfront induction chemotherapy"?

Title of Table 3, "front concurrent chemoradiotherapy...". Can we omit the "front" as it did not convey any meaning here?

Point by point response to Reviewer:

Reviewer #3:

The authors had tried very hard to address most of the comments. There is remaining concern on the methodology to assess the true treatment efficacy in locally recurrent NPC.

Despite all the above limitations, which were inherent in the interpretation of all studies in locally recurrent NPC, I still think that this study presented very interesting albeit preliminary findings in this difficult to treat situation where there is paucity of clinical trial results. Therefore, it would be important for the authors to highlight the above limitations in the interpretation of study results.

We thank the Reviewer for investing their time in carefully reading and commenting on our manuscript. These comments were very valuable and helped us to increase the clarity and quality of our study.

For this analysis, a total of 23 subjects were recruited within 12 months from June 2020 to May 2021 with data cut-off date in Dec 2021. The median follow up was less than 12 months for the entire cohort. With such a short follow up, it will be premature to make a definitive conclusion on the true efficacy of this combination approach.

REPLY: We thank the Reviewer for the very useful comment.

As we know, between June 1, 2020, and May 31, 2021, 23 patients were assigned to receive toripalimab plus capecitabine combination treatment. In our revised study, by the cut-off date of **Sep 1, 2023 (updated date)**, the median follow-up time was **29 months (IQR 26-33)**. As specified in the protocol, all enrolled patients will be followed up, and progressive disease (including local and distant failure) will be documented further. Then, the progression-free survival and duration of response may further confirm the benefit of toripalimab plus capecitabine in residual diseases (**Line 168 in page 9, line 221 in page 12, and lines 226-227 in page12**).

The location of the persistent or residual disease (Table 1) were at skull base (5/23, 21.7%) or skull base plus cervical lymph nodes (9/23, 39%), all within previously irradiated area.

First, it was well known that it was extremely difficult (if not impossible) by conventional imaging (MRI or CT scan, or even with PET-CT scan) to determine if any persistent or residual radiological abnormalities at skull base at 12-16 weeks post-RT to be post-RT inflammation/fibrosis vs genuine recurrence, unless there is

unequivocal radiological progression on serial follow up images.

REPLY: Thank you for the suggestion to improve the paper.

We agree that assessing tumour response using radiological imaging is difficult due to the difficulty of distinguishing post-treatment oedema, fibrosis, and necrosis from residual disease. As some studies revealed, different from edema, fibrosis and necrosis, the residual nasopharyngeal carcinoma can be diagnosed if the lesions remission (even accelerating remission) were observed by using medications (eg, toripalimab plus capecitabine) on serial MRI ¹. Therefore, we believed that point the Reviewer raised can be resolved by finding the unequivocal radiological remission on serial MRI in our study.

Secondly, for target lesion eligible for RECIST response criteria, tumor lesions situated in a previously irradiated area are usually not considered measurable unless there has been demonstrated progression in the lesion. As RECIST response rate was chosen as the primary endpoint of this trial, the inclusion of such RECIST non-measurable target lesions would raise uncertainty as to the validity of the exceptional high complete and partial response rate reported in this study. Progression free survival with adequate length of follow up would be a more clinically valid endpoint to assess the true efficacy of this combination for local recurrent diseases after prior chemoradiation.

REPLY: Thank you for the suggestion to improve our paper. We agree that it is important to briefly interpret the point, as this is the key policy assessed our work. In this trial, the residual nasopharyngeal carcinoma could not be identified and measurable on MRI were excluded. Furthermore, by using toripalimab plus capecitabine, 22 patients (95.7%) underwent unequivocal radiological remission on serial MRI. Based on these considerations, it is reasonable to believe that the primary endpoint is precise (Lines 102-103 in page 6, and protocol).

In our revised study, by the cut-off date of **Sep 1, 2023 (updated date)**, the median follow-up time was **29 months (IQR 26-33)**. As specified in the protocol, all enrolled patients will be followed up, and progressive disease (including local and distant failure) will be documented further. Then, the progression-free survival and duration of response may further confirm the benefit of toripalimab plus capecitabine in residual nasopharyngeal carcinoma (Line 168 in page 9, line 221 in page 12, and lines 226-227 in page12).

It was noted that pathological diagnoses were not obtained from most enrolled subjects and persistent or residual NPC was mainly diagnosed by imaging in this study. This is a well-recognized limitation on the interpretation of the efficacy result in locally recurrent NPC when the target lesions of interest were within previously irradiated area. It was well known that there is a high false positive rate for radiological abnormality in the skull base at the immediately post-RT period. The enrolment of subjects with false positive radiological abnormalities without histological confirmation or serial imaging to confirm true progression may have contributed to the falsely high complete response rate. Furthermore, nearly all

patients (95.7%) had negative plasma EBV DNA load at enrolment, further supporting that the suspicion that many of these radiological abnormalities may be false positive lesions that often regress or remain stable on serial follow up images.

REPLY: Thank you for reviewing our paper and providing constructive suggestions. We also appreciated the supportive suggestions.

In our revised study, in order to get an accurate diagnosis, the seriate sheet slices were constructed². Then, we have invited two senior pathologists and two cytologists to review the slices. As well as, we also invited two senior radiologists to review the images. According to the protocol, the residual nasopharyngeal carcinoma could not be identified and measurable on MRI in the area were excluded. Additionally, all of the patient without pathological or cytological confirmations underwent unequivocal radiological remission on serial MRI. After the Multi-Disciplinary Treatment (MDT), our results indicated that the pathological/cytological diagnoses, and radiological diagnoses was obtained in 10 patients and 13 patients, respectively. Based on these considerations, it is reasonable to believe that the primary endpoint is precise (**Lines 102-103 in page 6, and protocol**).

In our trial, the plasma EBV-DNA were detectable in two patients. The incidence of detectable plasma EBV-DNA was 8.7%, which was nearly consistent with the results in recent studies^{3,4}. Furthermore, it is known that the residual nasopharyngeal carcinoma can be diagnosed if the lesions were observed on the on serial MRI¹. In our trial, 22 patients (95.7%) underwent unequivocal radiological remission on serial MRI. Based on these considerations, we considered that the diagnoses of residual disease were rationally. It is reasonable to believe that the response rate is accurate.

It should also be acknowledged that plasma EBV DNA may not be detectable in most patient with small local recurrence. For reference, the sensitivity of EBV DNA by PCR assay was only up to 42.3% for the detection of local recurrence in a recent study. ([https://www.annalsofoncology.org/article/S0923-7534\(22\)00753-0/fulltext](https://www.annalsofoncology.org/article/S0923-7534(22)00753-0/fulltext))

REPLY: We agree that the plasma EBV-DNA may not be detectable in most small local recurrence. In our institution, for post-treatment EBV-DNA, **a cut-off value of 40 copies per mL was optimal for distinguish the two groups (detectable vs undetectable)**. Base on the consideration, the plasma EBV-DNA were detectable in two patients, and plasma EBV-DNA were undetectable in 21 patients (**Table 1**). In some circumstance, some patients with lower limit of plasma EBV-DNA (0-39 copies per mL) might potentially categorize into undetectable group. The incidence of detectable plasma EBV-DNA was 8.7% (2/23), which was nearly consistent with the results in recent reports^{3,4} (**Lines 330-335 in page 17**).

We thank the Reviewer for pointing out the important reference in the field. We are citing the reference when refer to the plasma EBV-DNA⁵.

Title of Table 2, "front induction chemotherapy". Should it be "upfront induction chemotherapy"?

REPLY: We thank the Reviewer for the very helpful correction. We have corrected the title of Table (**Lines 203-204 in page 11, and supplementary Table 1 in Supplement**).

Title of Table 3, "front concurrent chemoradiotherapy...". Can we omit the "front" as it did not convey any meaning here?

REPLY: Thank you for pointing out the error in table. We have corrected the title of Table ((**Lines 203-204 in page 11, supplementary Table 3 in Supplement**)).

References

- 1 Chen, Y. P. *et al.* Nasopharyngeal carcinoma. *Lancet*. **394**, 64-80 (2019).
- 2 Kononen, J. *et al.* Tissue microarrays for high-throughput molecular profiling of tumor specimens. *Nat Med*. **4**, 844-847 (1998).
- 3 Wang, W. Y. *et al.* Plasma Epstein-Barr virus DNA screening followed by (1)(8)F-fluoro-2-deoxy-D-glucose positron emission tomography in detecting posttreatment failures of nasopharyngeal carcinoma. *Cancer*. **117**, 4452-4459 (2011).
- 4 Lv, J. *et al.* Liquid biopsy tracking during sequential chemo-radiotherapy identifies distinct prognostic phenotypes in nasopharyngeal carcinoma. *Nat Commun*. **10**, 3941 (2019).
- 5 Chan, D. C. T. *et al.* Improved risk stratification of nasopharyngeal cancer by targeted sequencing of Epstein-Barr virus DNA in post-treatment plasma. *Ann Oncol*. **33**, 794-803 (2022).

REVIEWERS' COMMENTS

Reviewer #3 (Remarks to the Author):

The authors have satisfactorily addressed all the concerns raised.

Although the use of response rate as the primary endpoint for local recurrence in previously irradiated lesions could still be a major limitation, by the new updated data cut off of 1 Sep 2023, at a longer median follow-up time of 29 months, the PFS rates at 12- and 24-months were informative and provided further support for the clinical efficacy of this regimen.

Response to Reviewer:

Reviewer #3:

The authors have satisfactorily addressed all the concerns raised.

Although the use of response rate as the primary endpoint for local recurrence in previously irradiated lesions could still be a major limitation, by the new updated data cut off of 1 Sep 2023, at a longer median follow-up time of 29 months, the PFS rates at 12- and 24-months were informative and provided further support for the clinical efficacy of this regimen.

REPLY: We thank the Reviewer #3 for investing his/her time in carefully reading and commenting on our manuscript. These comments were very valuable and helped us to increase the clarity and rigor of our study.